# Transgenic Maize of *ZmMYB3R* Shapes Microbiome on Adaxial and Abaxial Surface of Leaves to Promote Disease Resistance

**DOI:** 10.3390/microorganisms13020362

**Published:** 2025-02-07

**Authors:** Shengqian Chao, Yin Zhang, Yue Hu, Yifan Chen, Peng Li, Yu Sun, Lili Song, Yingxiong Hu, Hui Wang, Jiandong Wu, Beibei Lv

**Affiliations:** 1Key Laboratory of Agricultural Genetics and Breeding, Biotechnology Research Institute, Shanghai Academy of Agricultural Sciences, Shanghai 201106, China; chaoshengqian@saas.sh.cn (S.C.); zhangyinsy1986@126.com (Y.Z.); huyue613@outlook.com (Y.H.); chenyifan@saas.sh.cn (Y.C.); sunsite@126.com (P.L.); yvessuen@hotmail.com (Y.S.); songlili@saas.sh.cn (L.S.); 2Key Laboratory for Safety Assessment (Environment) of Agricultural Genetically Modified Organisms, Ministry of Agriculture and Rural Affairs, Beijing 100125, China; 3Shanghai Professional Technology Service Platform of Agricultural Biosafety Evaluation and Testing, Shanghai 201106, China; 4CIMMYT—China Specialty Maize Research Center, Shanghai 201403, China; yxionghu@163.com (Y.H.); wanghui19840109@163.com (H.W.); 5National Engineering Laboratory of Crop Stress Resistance Breeding, School of Life Sciences, Anhui Agricultural University, Hefei 230036, China

**Keywords:** disease resistance, microbiome, niche, phyllosphere, transgene

## Abstract

The phyllosphere is one of the largest habitats for microorganisms, and host genetic factors play an important role during the interaction between microorganisms and the phyllosphere. Therefore, the transgene may also lead to changes in the maize phyllosphere. *ZmMYB3R* was identified as a drought-tolerant gene in *Arabisopsis*. Here, we employed metagenomic sequencing to analyze the microbiome of the adaxial and abaxial leaf surfaces on *ZmMYB3R*-overexpressing (OE) and wild-type (WT)·maize, aiming to dissect the possible associations between *ZmMYB3R* and changes in phyllosphere microbiome functioning. Our results revealed that overexpressing *ZmMYB3R* altered the alpha and beta diversity of the phyllosphere microbiome. In OE plants, more beneficial microbes accumulated on the phyllosphere, while pathogenic ones diminished, especially on the abaxial surface of *ZmMYB3R* leaves. Further analysis of disease resistance-related metabolic pathways and abundances of disease resistance genes revealed significant differences between OE and WT. The inoculation experiment between OE and WT proved that *ZmMYB3R* increased the disease resistance of maize. In conclusion, the results reveal that transgenes affect the phyllosphere microbiome, and *ZmMYB3R* might alter leaf disease resistance by reshaping the phyllosphere microbiome structure. These findings help us understand how *ZmMYB3R* regulates leaf disease resistance and may facilitate the development of disease control by harnessing beneficial microbial communities.

## 1. Introduction

The aboveground parts of plants are collectively referred to as the phyllosphere [1], which includes both vegetative organs (e.g., leaves and stems) and reproductive organs (e.g., flowers, fruits, and seeds) [2]. Unlike the internal environment of leaves, the environment on the leaf surface is unstable and complex, and the microbial structure of the phyllosphere is affected by many factors, such as ultraviolet radiation intensity, humidity, plant species, and genotypes [3,4]. The living environment of the microbiome on the leaf surface is harsh, as nutrients available for microbial utilization are limited and unequally distributed, and microbes also face the influence of plant defense mechanisms [5,6]. Nevertheless, various microorganisms are still found on the leaf surface of plants, including bacteria, fungi, and viruses [7,8].

Leaf characteristics such as leaf size, color, mineral content, the presence of leaf veins, stomata, and surface appendages (including trichomes and algal bodies) are also known to influence the composition of the microbial community [9,10]. Each plant leaf has two surfaces: the adaxial (upper) surface and the abaxial (lower) surface. The microbial community structure of the adaxial and abaxial surfaces of leaves may vary due to different environmental influences. The adaxial surface is usually exposed to strong light, with higher temperatures and abaxial humidity, while the abaxial surface is relatively cool and moist. These differences in the microenvironment lead to significantly distinct composition and function of the adaxial and abaxial surface microbial communities [10,11]. For example, microbes on the adaxial surface are often better adapted to high temperature and bright light conditions, while those on the abaxial surface have a preference for high humidity and low light conditions. Previous research demonstrated that the stomatal density on the abaxial surface of maize leaves was higher than that on the adaxial surface of maize leaves at both the flowering and the dough stages; the decrease in stomatal density reduced the transpiration and water availability on the surface of maize leaves, resulting in the decrease in microbial species diversity and population [12].

The phyllosphere microbiome can help plants fend off pathogen infection and tolerate abiotic stress; it also plays important roles in nutrient absorption and metabolic balance regulation [13]. It is known that the phyllosphere microbiome is influenced by complex interactions between the host, the microbiome, and the environment [14]. An increasing number of plant factors have been found to integrate biological/abiotic stress signals and actively participate in shaping the plant microbiome [15]. Northern corn leaf blight (NCLB) caused by *Exserohilum turcicum* is one of the most destructive maize fungal diseases in the world [16]. In recent years, studies have proved that maize varieties with different resistance to NCLB had different leaf chemical characteristics and unique phyllospheric bacterial communities [17]. Therefore, it may be feasible to control NCLB by maize leaf microbiome.

Drought-resistant transgenic maize exhibits stronger adaptability and survival ability under drought stress [18]. However, transgenes not only change the physiological characteristics of the host plant itself but may also have impacts on the environment. Genes encoding selective markers in transgenic plants may also disrupt the balance between commensal and pathogenic bacteria in transgenic plants [19]. Accordingly, altering the genotype of plants via transgenic means may change the distribution of the microbiome in the phyllosphere. In turn, this disturbance in the microbiome on the adaxial and abaxial surfaces of leaves may affect leaf function. Previous research found that the maize *ZmMYB3R* gene may confer drought tolerance to plants under drought conditions, and the overexpression of this gene in *Arabidopsis* increased stomatal closure and the abscisic acid content and reduced reactive oxygen species accumulation through promoting the activities of superoxide dismutase and catalase [20]. It is worth noting that these phenotypic changes may also alter water content and nutrient secretion on the leaf surface, and, therefore, microbial growth and colonization would be modified, which might induce the change of function of the phyllosphere microbiome.

In the present study, microbiome samples of the adaxial and abaxial leaf surfaces of transgenic maize overexpressing *ZmMYB3R* and wild-type (WT) maize were collected. Metagenomic sequencing was employed to comprehensively analyze the composition and function of the phyllosphere microbial community. This technique not only provides information regarding the diversity and abundance of the microbial community but also may reveal the distribution and metabolic potential of functional genes in the microbial community. Microbial communities on the adaxial and abaxial surfaces of the leaves of *ZmMYB3R*-overexpressing transgenic maize (OE) and non-transgenic maize (WT) were compared to explore the effects of transgenes on plant–microbial interactions and uncover the potential ecological impacts. This study can help assess the ecological risks of transgenic crops, identify beneficial microbial communities for biological disease resistance, and facilitate the development and management of transgenic crops in the future.

## 2. Materials and Methods

### 2.1. Experimental Design and Sampling

A pot experiment was performed in a greenhouse (25–28 °C, 16 h light/8 h darkness). The seeds of transgenic maize overexpressing *ZmMYB3R* (OE) and the wild-type line (WT) were provided by the School of Life Sciences, Anhui Agricultural University, and were sown in soil collected from the Baihe base (latitude 31°24′ N and longitude 121°11′ E). Three replicates for each group and three seedlings for each replicate. Seedlings were watered once or twice per week, depending on soil humidity. Adaxial and abaxial leaf surfaces of plants at the seedling stage (V6) were sampled for the metagenomic sequencing experiment, and the samples were named WT-adaxial, OE-adaxial, WT-abaxial, and OE-abaxial. The sampling time and position of several samples were the same. The leaves were the fifth and sixth leaves counted from bottom to top. The two surfaces of the leaves were from the same position on the same leaf of the same seedling. The sampling position on the adaxial surface of the leaves corresponded to the sampling position on the abaxial surface.

### 2.2. Sample Collection

During sampling, microorganisms on 10 g of plant leaves were randomly scraped from the same position and area for both the adaxial and abaxial surfaces of the leaves using sterile cotton swabs and mixed with 90 ml of sterile 0.1 mmol/L potassium phosphate buffer solution (pH = 7.4). The samples were shaken for 5 min using a laboratory shaker, sonicated for 1 min, and vortexed for 10 sec. The above procedure was repeated twice, and then the washing solution was passed through a 0.22 μm filter membrane [12,21]. The used filter membrane was frozen in liquid nitrogen and stored at −80 °C.

### 2.3. DNA Extraction and Sequencing

Total microbial genomic DNA samples were extracted using the OMEGA Mag-Bind Soil DNA Kit (M5635–02) (Omega Bio-Tek, Norcross, GA, USA) following the manufacturer’s instructions and stored at −20 °C prior to further assessment. The quantity and quality of extracted DNA were measured using a Qubit™ 4 Fluorometer (Qubit™ Assay Tubes: Q32856; Qubit™ 1X dsDNA HS Assay Kit: Q33231) (Invitrogen, Waltham, MA, USA) and agarose gel electrophoresis. The extracted microbial DNA was processed to construct metagenome shotgun sequencing libraries with insert sizes of ~400 bp using the Illumina TruSeq Nano DNA LT Library Preparation Kit (Illumina, San Diego, CA, USA). Each library was sequenced on the Illumina NovaSeq platform (Illumina, San Diego, CA, USA) according to the PE150 strategy at Personal Biotechnology Co., Ltd. (Shanghai, China). All raw sequences were deposited in the National Center for Biotechnology Information (NCBI) Sequence Read Archive under accession number PRJNA1138799.

### 2.4. Bioinformatics Analyses of the Metagenomes

Community diversity analysis was performed using the QIIME 2 platform, including the calculation of the alpha diversity (Shannon index and Chao1 index) and the beta diversity (Bray–Curtis distance). Mothur software (v.1.48.2) was employed to calculate the Spearman rank correlation coefficients between the bacterial, fungal, and metabolomics data and the microbial abundance. A correlation network was constructed for the relevant information with |rho| > 0.8 and *p* value < 0.01 and imported into Cytoscape software (v.3.10.3) for visualization. To assess the metabolic potential of the microbial populations, the results of functional annotation were utilized for pathway analysis using the Comprehensive Antibiotic Resistance Database (CARD) and the Kyoto Encyclopedia of Genes and Genomes (KEGG) database. All graphing and difference testing were implemented in R (version 3.6).

### 2.5. Exserohilum Turcicum Inoculation and Assessment of Plant Disease Resistance

We used lab-preserved *E. turcicum* strains, which were transferred from glycerol stored at −80 °C to PDA medium and cultured at room temperature for 2–3 weeks. The spore suspension of *E. turcicum* was prepared by flooding each PDA plate with approximately 8 mL of ddH_2_O, which contained 0.1% Tween 20. The spores were then gently removed using a glass rod. Subsequently, approximately 5 mL of the undiluted spore suspension was transferred into each centrifuge tube. To adjust the concentration of the suspension, additional ddH_2_O was added until it reached a final concentration of 4 × 10^3^ conidia per ml [22]. For the greenhouse experiment, to investigate the resistance of WT and *ZmMYB3R*-OE maize to *E. turcicum*, an artificial inoculation procedure was carried out as described by Wu et al. [23]. Fourteen days post inoculation, leaf samples were collected from the inoculated area with visual symptoms. Disease severity was evaluated in nine plants based on a diagrammatic scale [24].

## 3. Results

### 3.1. Structural Changes of the Phyllosphere Microbiome Induced by Overexpressing ZmMYB3R

The sequence analysis showed the average sequence numbers of WT-adaxial, OE-adaxial, WT-abaxial, and OE-abaxial (Appendix A). *ZmMYB3R*-OE maize exhibited significantly reduced bacterial (*p* = 0.001) and fungal (*p* = 0.001) Chao1 index values (Figure 1a,b) compared to WT. According to beta diversity, genotype (*p* = 0.001 for both bacteria and fungi) significantly altered the structure of bacterial and fungal communities (Figure 1c,d). The dominant phylum of the phyllosphere microbiome of WT and OE were similar (Figure 1e,f), both with *Proteobacteria*, *Actinobacteria*, and *Bacteroidota* as the dominant bacterial phyla and *Basidiomycota*, *Ascomycota*, and *Mucoromycota* as the dominant fungal phyla. However, the relative abundances of these microbes differed in these two genotypes (Figure 1e,f).

The effects of the transgene (*ZmMYB3R*-OE) on microbes on the adaxial and abaxial surfaces of leaves were further analyzed. *ZmMYB3R*-OE significantly influenced the bacterial and fungal Shannon diversity and Chao1 index values (Table 1), while niche only significantly influenced the fungal Shannon diversity (*p* = 0.037) and Chao1 index (*p* = 0.005) (Table 1), indicating that the effect of the transgene on the microbiome was greater than that of the niche. Specifically, *ZmMYB3R* reduced the bacterial and fungal Chao1 indices for both surfaces of leaves, and the difference was not significant between the two genotypes (Appendix A). The bacterial and fungal Chao1 index values of the adaxial surfaces of leaves were higher than those of the abaxial leaf surfaces. Interestingly, the overexpression of *ZmMYB3R* reduced the difference in the Chao1 index between the adaxial and abaxial surfaces of leaves. Regarding beta diversity, the structure of bacterial and fungal communities in WT maize was not significantly altered by the niche (*p* = 0.001 for both bacteria and fungi) (Appendix A; Table 2). In addition, the WT and OE maize harbored significantly different bacterial and fungal communities in different ecological niches (Table 2, pairwise comparison). Principal component analysis (PCA) of the identified microbial phyla, as indicated in Figure 1, revealed that WT maize had the largest dispersion of the phyllosphere microbial community (Appendix A).

Similar community compositions were found between the phyllosphere communities of the two genotypes among different niches (Appendix A). Specifically, the dominant bacterial phyla were *Proteobacteria*, *Actinobacteria*, and *Bacteroidota*, with relative abundance ranges of 69.24–88.81%, 12.12–4.39%, and 0.79–10.69%, respectively (Appendix A). The fungal communities were mainly composed of *Basidiomycota*, *Ascomycota*, and *Mucoromycota*, and their relative abundances were 37.12–77.24%, 48.99–20.23%, and 13.10–1.46%, respectively (Appendix A). Seven bacterial phyla (*Bacteroidota*, *Acidobacteria*, *Chloroflexi*, *Proteobacteria*, *Armatimonadetes*, *Deinococcus-Thermus*, and *Actinobacteria*) and three fungal phyla (*Mucoromycota*, *Basidiomycota*, and *Ascomycota*) were significantly influenced by niches (Appendix A). The relative abundance of *Basidiomycota* was higher on the abaxial surface of leaves, while *Mucoromycota* exhibited the opposite (Appendix A). Two bacterial phyla (*Proteobacteria* and *Bacteroidota*) and three fungal phyla (*Basidiomycota*, *Ascomycota*, and *Mucoromycota*) may have been influenced by the transgene (Appendix A). Of these phyla, the relative abundances of *Proteobacteria* and *Basidiomycota* were higher in *ZmMYB3R*-OE maize, while the other phyla showed the opposite pattern. Only one bacterial phylum (*Cyanobacteria*) was significantly influenced by the interactive effects of the niche and transgene (Appendix A). It was noteworthy that *Cyanobacteria* bacteria were significantly more abundant in WT-adaxial samples compared to the other samples. Further analysis revealed that two bacterial species (*Burkholderia gladioli* and *Stutzerimonas stutzeri*) and three fungal species (*Bipolaris sorokiniana*, *Rhizopus arrhizus*, and *Bipolaris maydis*) were significantly influenced by the niche (Appendix A).

### 3.2. Overexpression of ZmMYB3R Altered the Stability of the Phyllosphere Microbiome Co-Occurrence Network

A microbial co-occurrence network was constructed, and the complexity of the network connectivity of the two leaf surfaces of WT and OE maize was analyzed (Figure 2; Table 3). Overall, *Ascomycota* was the most abundant phylum in four types of samples (WT-adaxial, WT-abaxial, OE-adaxial, and OE-abaxial), and the network structure of the phyllosphere was significantly influenced by trans-*ZmMYB3R*, yet the responses of the two leaf surfaces to trans-*ZmMYB3R* were different. The negative correlation and negative/positive values of WT-adaxial were higher than that of WT-abaxial, which demonstrated that the network of microbial communities on abaxial surfaces (the proportion of negative edges/average degree: 51.25%/17.48) was not stable as microbial community network on the adaxial surfaces (the proportion of negative edges/average degree: 52.2%/17.8), while the opposite pattern was found for OE maize (Table 3). In OE-abaxial, the negative correlation and negative/positive values were much higher than those of OE-adaxial, WT-adaxial, and WT-abaxial, reflecting a more complex and robust co-occurrence network. Trans-*ZmMYB3R* enlarged the difference in stability of the microbial networks on the abaxial surfaces of leaves. Interestingly, the ratios of fungal/bacterial nodes for the adaxial surfaces were higher than those for the abaxial surfaces in both WT and OE maize, demonstrating that the interaction between bacteria and fungi differed between the two leaf surfaces (Table 3).

### 3.3. Overexpression of ZmMYB3R Changed the Bacterial and Fungal Species of the Adaxial and Abaxial Surfaces of Leaves in WT and OE Maize

To identify the differences in bacterial and fungal communities on the two leaf surfaces of OE and WT maize, 12 dominant (relative abundance ≥ 1.0%) bacterial and fungal groups at the species level were detected (Figure 3), and analysis of variance (ANOVA) was performed. These results showed that the dominant species category was the same in different samples (Figure 3a,b). *Rugosimonospora africana* was slightly more abundant in WT-adaxial than in OE-adaxial, WT-abaxial, and OE-abaxial samples (*p* < 0.001) (Figure 3c). *Tilletiaria anomala* was slightly more abundant in OE-abaxial than in OE-adaxial, WT-adaxial, and WT-abaxial samples (Figure 3d). *Bipolaris maydis* and *Rhizopus arrhizus* were more abundant in WT-adaxial than in OE-adaxial and OE-abaxial samples (Figure 3d). *Bipolaris sorokiniana* was more abundant in WT-adaxial and WT-abaxial samples than in OE-adaxial and OE-abaxial samples (*p* < 0.001, ANOVA) (Figure 3d).

Overall, there were more beneficial microorganisms (such as *Burkholderiales bacterium*, *Burkholderia gladioli*, *Methylorubrum zatmanii*, *Methylobacterium aquaticum*, and *Sphingomonas* sp. *MA1305*) and fewer pathogens (such as *Bipolaris maydis*, *Rhizopus arrhizus*, *Bipolaris sorokiniana*, *Moesziomyces antarcticus*, *Moesziomyces aphidis*, *Hortaea werneckii*, *Ustilago trichophora*, *Venturia nashicola*, and *Pyricularia oryzae*) in the OE-abaxial community (Figure 3). Taken together, these results indicate that OE maize is affected by fewer pathogens than WT maize, possibly leading to promoted disease resistance.

### 3.4. OE Maize ZmMYB3R Showed No Difference in the Dominant Resistance Gene Category

To investigate whether the disease resistance of different maize genotypes changed, we screened the CARD and analyzed the antibiotic-resistant genes of four types of samples (WT-adaxial, WT-abaxial, OE-adaxial, and OE-abaxial) (Appendix A). The top 15 most abundant genes in the phyllosphere of the two genotypes together accounted for 31.07% for WT-adaxial, 30.98% for WT-adaxial, 31.63% for OE-adaxial, and 33.27% for OE-abaxial in terms of relative abundance (Figure 4a–d), and these genes were the same. The most abundant functional gene was *Ecol_fabG_TRC* (6.55–6.95% relative abundance, Figure 4a–d). *Ecol_fabG_TRC* is involved in the resistance mechanism of antibiotic target alteration. The results showed that other highly abundant functional genes were involved in antibiotic efflux, resistance by absence, and antibiotic inactivation.

In contrast to the niche, the transgene *ZmMYB3R* did significantly affect the functional gene profile of microbial resistance in the phyllosphere (Figure 4e). The functional gene profile of the microbiome between WT and OE was also analyzed (Appendix A). The abundance of *Saur_parC_FLO*, *Cgin_gyrA_FLO,* and *mecA* in OE was lower than WT, while *Bbac_gyrA_FLO* was higher than WT. *Saur_parC_FLO*, *Cgin_gyrA_FL,* and *Bbac_gyrA_FLO* were all involved in the resistance mechanism of antibiotic target alteration, and *mecA* was involved in antibiotic target replacement. Detailed analysis showed that the relative abundance of *Cgin_gyrA_FLO* was significantly different in WT-adaxial, WT-abaxial, OE-adaxial, and OE-abaxial samples (*p* < 0.001, ANOVA, Figure 4f). Its abundance in WT-adaxial and WT-abaxial were both higher than in OE-adaxial or OE-abaxial. These results showed that the transgene *ZmMYB3R* changed the microbiome abundance difference between the adaxial and abaxial surfaces of leaves, which further leads to differences in disease resistance between the adaxial and abaxial surfaces of leaves.

### 3.5. Overexpressing ZmMYB3R Changed the Abundance of the Functional Microbiome and Genes in the Maize Phyllosphere

To further investigate the possible role of the core microbiome in disease resistance, the functional genes were analyzed using the KEGG database and the CARD resistance gene database (Figure 5a). A total of nine disease resistance pathways were enriched, namely, ko00280, ko00940, ko01501, ko01502, ko01503, ko02020, ko02024, ko04075, and ko04626, which were involved in aromatic compound metabolism, phenylpropanoid biosynthesis, beta-lactam resistance, vancomycin resistance, cationic antimicrobial peptide (CAMP) resistance, two-component systems, quorum sensing, plant hormone signal transduction (jasmonic acid and ethylene signaling pathway), and plant–pathogen interactions, respectively. More genes were enriched in most disease resistance-related pathways in OE-abaxial samples than in WT-abaxial and WT-adaxial samples (Appendix A), indicating higher disease resistance in OE-abaxial samples than in WT maize. Interestingly, for the ko01502 pathway, the gene enrichment in WT was higher than that in OE, and the OE-abaxial samples exhibited almost no enrichment in this pathway.

*Cgin_gyrA_FLO* is involved in the resistance mechanism of antimicrobial target alteration, which is the same mechanism corresponding to ko01503. In addition, it was found that *mecA* is a β-lactam resistance gene belonging to ko01501. The relative abundances of *Cgin_gyrA_FLO* and *mecA* in the microbial communities of the four samples (WT-adaxial, WT-abaxial, OE-adaxial, and OE-abaxial) presented significant differences (Figure 5b,c). Compared with WT maize, *Chryseobacterium* in OE maize had an absolute advantage in its contribution to *Cgin_gyrA_FLO*. *Sediminibacterium roseum* made a great contribution to OE-adaxial. *Rudanella lutea*, *Arsenicibacter rosenii*, and *Rudanella paleaurantiibacter* contributed predominately to *Cgin_gyrA_FLO* in WT maize. For *mecA*, the contribution of the microbiome differed in all four samples. *Sphingomonadales* contributed the most in OE-abaxial; *Massilia* and *Cutibacterium* acnes contributed the most in OE-adaxial; *Bacillus*, *Microbacterium*, and *Exiguobacterium* contributed the most in WT-abaxial; and *Cyanobacteria* contributed the most in WT-adaxial. This result suggested that the different core microbiomes on the adaxial and abaxial surfaces of leaves lead to the differential abundance of disease resistance genes, giving rise to varying levels of disease resistance.

### 3.6. Overexpressing ZmMYB3R Improves Resistance to Exserohilum Turcicum in Maize

In order to further investigate whether *ZmMYB3R* changes the disease resistance of maize, we inoculated maize with the pathogen *Exserohilum turcicum* at the V4 stage (Figure 6a). *Exserohilum turcicum* belongs to *Ascomycota*, and the disease symptoms of maize were mostly characterized by yellowing, severe necrosis, and strong leaf blight, which are similar to the symptoms of *Bipolaris maydis*. Fourteen days after inoculation, the disease severity of WT was significantly higher than OE. WT presented typical brown necrotic spots, spreading along the vein, grayish brown or yellowish brown, mostly long fusiform, and the difference in symptoms between WT and OE was evident (Figure 6b). These results showed that *ZmMYB3R* increased the resistance to *Exserohilum turcicum* in maize.

## 4. Discussion

Plants are colonized by highly diverse and tissue-specific microbial communities collectively termed plant microbiota [25]. Similar to roots, microbial communities, including bacteria and fungi, also colonize the surface of leaves [26]. In the phyllosphere, the microbial community has the potential to promote the tolerance of host plants to various abiotic and biotic stresses [27]. The phyllosphere microbiome is easily affected by various factors, and the microbiome on the adaxial and abaxial surfaces of leaves differs [15]. To date, many studies have explored the impact of transgenes on the diversity of plant microbial communities. For example, the transgene of *BAR11* increased the relative abundance and diversity of rhizosphere microbial communities [28]. In contrast, relatively few studies have reported the impact of transgenes on the community diversity of the phyllosphere microbiome. Previous research found that the alpha diversity of the Bt cotton strain *SGK321* phyllosphere microbiome was lower than that of the control [29]. Little research has examined the effects of transgenes on the microbiome of the adaxial and abaxial surfaces of leaves.

This study investigated the effects of niche and transgenes on the maize phyllosphere microbiome using metagenomic sequencing. Profiling the microbiome on the adaxial and abaxial surfaces of leaves of WT and *ZmMYB3R*-OE plants revealed that the *ZmMYB3R* transgene and the niche significantly affected the composition of the phyllosphere microbiome (Figure 1 and Appendix A). These findings provide evidence that the niche and transgenes not only can impact the diversity and assembly of phyllosphere microbial communities but also may affect their ecological functions. In the following sections, we discuss how these findings advance the understanding of transgene-induced changes in plant microbiome assembly, co-occurrence patterns, and functions.

Previous studies have shown that the phyllosphere microbiome of plants is mainly actinomycetes, Bacteroides, Firmicutes, and Proteobacteria [30]. In our study, these bacteria were all found in the phyllosphere bacterial communities of four types of samples (Figure 1e–f and Appendix A), suggesting that these bacteria are common in the plant phyllosphere. Water availability on the surface of maize leaves affects the microbial phyllosphere composition [12]. *ZmMYB3R* alters ABA content and stomatal closure in *Arabidopsis*. Therefore, it was speculated that the structure and function of the microbial community on the adaxial and abaxial leaf surfaces would also change with the overexpression of *ZmMYB3R* in maize. Indeed, in this study, we found that transgene *ZmMYB3R* significantly affected the composition of phyllosphere microbial communities. The overexpression of *ZmMYB3R* in maize changed the alpha and beta diversity, demonstrating that *ZmMYB3R* had a significant effect on bacterial and fungal communities in the phyllosphere (Figure 1 and Appendix A, Table 1). However, a significant niche ×*ZmMYB3R* interaction was also observed in the fungal community (Table 2). The OE-abaxial Chao1 index was the lowest, and there were more *Proteobacteria* and fewer *Mucoromycota* in the OE-abaxial community (Appendix A). Several studies analyzing bacterial diversity in different niches have provided evidence that changes in microbial community composition are the foundation for maintaining host health and disease status [31,32]. In this study, the co-occurrence network of OE-abaxial was the most stable in four types of samples (Figure 2). Studies have shown that the rhizosphere microbiome of the low-cadmium rice XS14 has a stable symbiotic network and strong resistance to changes in soil properties [33]. Drought promotes the unstable characteristics of soil bacterial symbiotic networks [34]; therefore, it was speculated that the OE-adaxial microbial community was more tolerant to external environmental changes and both abiotic and biological factors.

Further analysis of the microbial species on the adaxial and abaxial leaf surfaces of WT and OE maize revealed that there was a more beneficial microbiome in the OE-abaxial community (Figure 3). In addition, there were fewer pathogens in the WT-abaxial community than in the WT-adaxial community (Figure 3). Changes in the composition of the microbiome may improve disease resistance by enriching beneficial microorganisms and enhancing the plant’s immune system. Transgenic crops may promote the growth of beneficial microorganisms by changing the chemical composition of the soil or rhizosphere environment and enhancing the plant’s immune system through the expression of disease-resistant genes [30,35]. *Burkholderia* is an important group of beneficial microorganisms associated with plants [36]. *Burkholderia* can secrete organic acids or acid phosphatases to convert insoluble phosphorus in the soil into soluble phosphorus that can be directly absorbed and utilized by plants [37], degrade pollutants in the environment [38], and produce plant hormones, such as indole acetic acid [39] and secrete antibacterial substances, to inhibit the growth of pathogenic bacteria [40]. For example, *Burkholderia cepacia MCI7* can antagonize and inhibit major fungal pathogens in maize soil [41]. *B. Ambifaria* inhibits the growth of various pathogenic fungal hyphae in soil by releasing diffusible antibacterial compounds. This strain, as a seed coating agent, can significantly inhibit the infection of pathogenic bacteria (*Pythium ultimum*) on cucumber and soybean, maintaining the health of cucumber and soybean [42]. *CF66I*, an antifungal compound produced by *Burkholderia cepacia*, has been reported to have inhibitory effects on the growth of a variety of pathogenic fungi [43]. Due to these benefits, *Burkholderia* species are commonly employed for biological control. *Methylorubrum* species can be utilized to reduce environmental pollution because of their ability to degrade toxic compounds, tolerate high concentrations of heavy metals, and increase plant tolerance to these compounds [44]. There were some strains of *Methylobacterium*/*Methylorubrum*, even at low abundance, leading to increased growth of inoculated plants [45]. This function is prevalent in *Methylobacterium*/*Methylorubrum* strains. *Sphingomonas* has numerous functions, from repairing environmental pollution to producing highly beneficial plant hormones and improving plant growth under stress conditions such as drought, salinity, and heavy metals in agricultural soils, attributed to their potential production of plant growth hormones, such as gibberellin and indole-3-acetic acid [46].

In many plant species, both surfaces of the leaf may be susceptible to adaptive powdery mildew fungi, yet in some plant species, including *Arabidopsis*, immunity on the abaxial surface of the leaf has been observed. The genetic basis of this dorsal leaf immunity is unknown [47]. Although not tested in this study, our results suggest that transgenic *ZmMYB3R* may improve the disease resistance of the phyllosphere, especially on the abaxial surface of leaves. However, the mechanism through which *ZmMYB3R* enhances leaf disease resistance remains to be determined, and further detailed studies based on in-depth comparison of the functional microbiome of the WT and OE phyllosphere are necessary.

Consistent with the microbial community response, significant and consistent changes were observed in the relative abundances of specific disease resistance microbial functional gene categories and related metabolic pathways in the phyllosphere of OE maize compared to WT maize. Changes in the abundance of these functional genes may indicate changes in leaf-associated functions related to disease resistance. The composition of phyllosphere microbial communities is associated with temperature, humidity, nutrients, CO_2_, and genotype [3,4,8]. The temperature and light intensity differences between the two surfaces of leaves may also affect the severity of diseases caused by bacterial and fungal pathogens [48]. While we did not analyze the water content of WT-adaxial, WT-abaxial, OE-adaxial, and OE-abaxial samples, *ZmMYB3R* has been shown to enhance drought resistance by controlling increased stomatal closure and abscisic acid (ABA) content in *Arabidopsis* [20]; the opening and closing of stomata can control the gas exchange and water evaporation between plants and the outside world. There may be anatomical differences in stomatal density on the surface of two leaves; the stomata number, stomata index, and stomata length increased in the upper surfaces of leaves in comparison with their lower surfaces [49], and stomatal density is positively correlated with the development of diseases [50]. The local relative humidity on the abaxial of leaves is usually higher than that on the adaxial, which may promote the occurrence of downy mildew caused by oomycete pathogens [51,52]. In addition, although the analysis using the CARD showed no difference in dominant gene categories (Figure 4a–d), there were differences in the abundances of some disease resistance genes and genes involved in resistance metabolic pathways (Figure 4f and Figure 5). Hence, the results confirm the importance of the genotype in determining microbial composition.

The metagenomic sequencing results analyzed disease resistance pathways that were enriched by functional genes. Changes in plant metabolic pathways and microbial metabolic pathways may lead to changes in the balance between symbiotic microorganisms and pathogenic microorganisms. Ko00940 was related to phenylpropanoid biosynthesis, and this pathway generates many secondary metabolites related to disease resistance. The secretion of secondary metabolites, such as phenolic acids secreted by plants, has antibacterial effects and can inhibit the reproduction of certain pathogenic microorganisms [53]. Sugars and amino acids secreted by plants can provide nutrients for nitrogen-fixing bacteria or growth-promoting bacteria, helping them to colonize the rhizosphere [53]. Plant hormones not only regulate the growth of plants themselves but also affect their interactions with plants by regulating the composition of microbial communities. Auxins and auxin-producing bacteria can promote the production of antioxidant compounds and osmotic regulators in plants, which can counteract cadmium toxicity, promote plant growth, and improve plant remediation efficiency [54]. Auxins, cytokinin, and ethylene can affect rhizosphere microorganisms and induce systemic drought tolerance in plants [55]. ABA affects the rhizosphere bacterial community during the later developmental stages, thereby further influencing the composition of rhizobia by affecting compounds in root exudates [56]. Ko04075 was a plant hormone signal transduction, although the abundance was low, and *ZmMYB3R* has been shown to enhance drought resistance by controlling increased stomatal closure and abscisic acid (ABA) content in *Arabidopsis* [20]. The evolutionary trajectories of symbiotic and pathogenic interactions are constrained and interconnected. Symbiotic microorganisms usually inhibit the growth of pathogenic microorganisms by synthesizing antibiotics, enzymes, or toxic metabolites [57]. Changes in the host immune system (such as by initiating innate immune responses, changing the distribution or function of immune cells, etc.) may affect the competition and symbiotic relationship between symbiotic microorganisms and pathogenic microorganisms, thereby affecting host health and disease resistance [53].

The relative abundances of most resistance-related metabolic pathways in the OE-abaxial community were higher than in the WT-adaxial, WT-abaxial, and OE-adaxial communities, especially ko02020 and ko02024 (Appendix A). Two-component systems (ko02020) are extensively studied in microbiology as essential signaling mechanisms utilized by bacteria to adapt to changing environments, including host–pathogen interactions [58,59]. Microbial pathogenicity studies have established that quorum sensing (ko02024) regulates the Type III secretion system (T3SS) in bacteria, particularly in pathogenic strains. Quorum sensing involves bacteria sensing their population density through signaling molecules, such as acyl-homoserine lactones (AHLs), which in turn can regulate virulence mechanisms, including the T3SS [60]. Bacterial pathogens, such as *Pseudomonas syringae,* use quorum sensing to regulate the secretion of effector proteins via the T3SS, depending on the local bacterial population density [61,62]. Little genes in the OE-abaxial community were enriched to the vancomycin resistance pathway (ko01502). Vancomycin (VCM), a glycopeptide antibiotic that can inhibit the synthesis of peptidoglycans during bacterial cell wall production, is commonly detected in research on bacterial resistance in human diseases [63].

The relative abundances of *cgin_gyrA_FLO* and *mecA* were higher in WT maize compared to OE maize (Figure 4f and Appendix A). *cgin_gyrA_FLO* and *mecA* are involved in cationic antimicrobial peptide (CAMP) resistance (ko01503) and beta-lactam resistance (ko01501), respectively. Both pathways are related to the mechanism of microbial resistance to antibiotics [64,65]. Further species contribution analysis showed that the contributions of different microorganisms to *cgin_gyrA_FLO* and *mecA* varied in the OE-abaxial, OE-adaxial, WT-abaxial, and WT-adaxial communities (Figure 5b,c), which is likely to be one of the factors leading to the different levels of disease resistance. Because most of these microorganisms have biocontrol functions, for example, *Bacillus*, it was found that the *Bacillus* that strongly inhibits maize NCLB was from the maize epiphytic bacteria groups and showed positive resistance to its pathogens [66]. It is well known that plants can specifically recruit beneficial microbes that induce disease resistance and promote growth [67]. *Arabidopsis* increases malic acid secreted by the roots to attract *Bacillus subtilis* around the roots, resulting in a stronger host response to *Pseudomonas syringae* [68]. These results indicate that the response of the phyllosphere microbial genes plays an important role in disease resistance metabolism after the transfer of *ZmMYB3R*. The inoculation experiment of *Exserohilum turcicum* further proved that the transfer of *ZmMYB3R* did improve the disease resistance of maize (Figure 6). However, due to the complexity of the disease resistance mechanism, further pathogen inoculation and molecular experiments on adaxial and abaxial surfaces of leaves are necessary to investigate the effect of *ZmMYB3R* on maize phyllosphere disease resistance.

The overexpression of *ZmMYB3R* may help to regulate the microbial community on the phyllosphere, selectively promote the growth of beneficial microorganisms, and inhibit the expansion of pathogenic microorganisms. The transgenic maize enhanced the maize immune response, especially against common diseases (such as NCLB (northern corn leaf blight)), thereby improving maize disease resistance. The result also provides a theoretical basis for future agricultural production. Growers can consider cultivating maize varieties that overexpress *ZmMYB3R* and use biological control methods, such as probiotics or biopesticides, to further enhance the health of leaf microbial communities. Based on the impact of *ZmMYB3R* on the microbial community, maize growers can improve soil health and promote the colonization of beneficial microorganisms by increasing the content of soil organic matter and adjusting soil pH. Reasonable irrigation and fertilization can maintain the stability of phyllosphere and rhizosphere microbial communities and avoid overfertilization and waterlogging, which may affect the balance of microorganisms and thus affect disease resistance. When planting maize, it can be rotated with other crops (such as beans and wheat) to change the structure of the microbial community, reduce the accumulation of soil pathogens, and promote the reproduction of beneficial microorganisms in the soil, thereby indirectly improving the disease resistance of maize.

## 5. Conclusions

Transgenic varieties have become increasingly common in our current agriculture. However, many genes are pleiotropic, and unexpected effects should be carefully examined when new genes are transferred. Metagenomic sequencing of the phyllosphere microbiome suggested that *ZmMYB3R* might improve the disease resistance of maize. The transgenic overexpression of *ZmMYB3R* in maize altered the phyllosphere microbiome composition and the difference between the adaxial and abaxial surfaces of leaves. The microbiome of OE-abaxial samples contained more beneficial microbes and fewer pathogens compared to the other types of samples (OE-adaxial, WT-abaxial, and WT-adaxial), and the co-occurrence network was more stable than that for other niches. Functional gene profiling revealed that the relative abundances of resistance genes and disease resistance metabolic pathways were different in OE-abaxial, OE-adaxial, WT-abaxial, and WT-adaxial. The inoculation experiment proved that *ZmMYB3R* increased the maize phyllosphere disease resistance. Thus, the results of this study may be helpful for understanding how *ZmMYB3R* uses the microbiome to influence the disease resistance mechanism of maize leaves, supporting the potential use of a beneficial microbiome for the biological control of diseases.

## Figures and Tables

**Figure 1 microorganisms-13-00362-f001:**
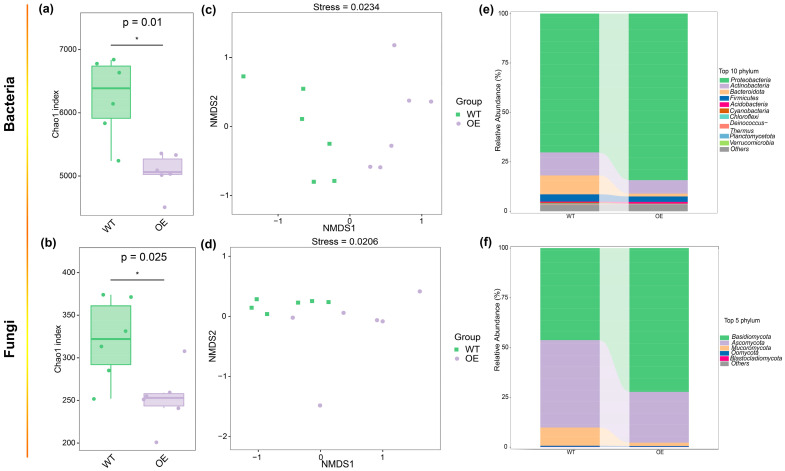
Bacterial and fungal communities in tested maize samples. Chao1 index values of (**a**) bacterial and (**b**) fungal communities in the maize phyllosphere (one-way analysis of variance (ANOVA), *n* = 3, *p* < 0.05). Non-metric multi-dimensional scaling (NMDS) ordinations based on the Bray–Curtis similarity showed differences in the structure of (**c**) bacterial and (**d**) fungal communities in the phyllosphere of wild-type (WT) and *ZmMYB3R*-overexpressing (OE) maize (permutational ANOVA (PERMANOVA), *n* = 3, *p* < 0.05). Relative abundances of (**e**) bacterial and (**f**) fungal phyla. OE represents transgenic maize overexpressing *ZmMYB3R* and WT represents wild-type maize. Significant differences are marked with asterisks: *, *p* < 0.05.

**Figure 2 microorganisms-13-00362-f002:**
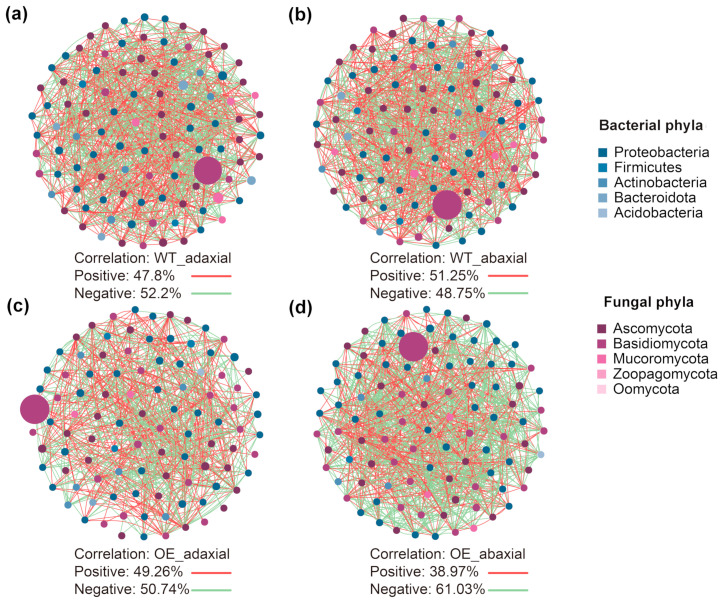
Co-occurrence network of the phyllosphere microbial community for (**a**) WT-adaxial, (**b**) WT-abaxial, (**c**) OE-adaxial, and (**d**) OE-abaxial samples. Blue nodes represent bacteria, violet nodes represent fungi, green lines represent negative correlations, and red lines represent positive correlations. The node size indicates the degree of connection. WT-adaxial, adaxial surfaces of leaves from wild-type maize; OE-adaxial, adaxial surfaces of leaves from transgenic maize (*ZmMYB3R*-overexpressing maize); WT-abaxial, abaxial surfaces of leaves from wild-type maize; OE-abaxial, abaxial surfaces of leaves from *ZmMYB3R*-overexpressing maize.

**Figure 3 microorganisms-13-00362-f003:**
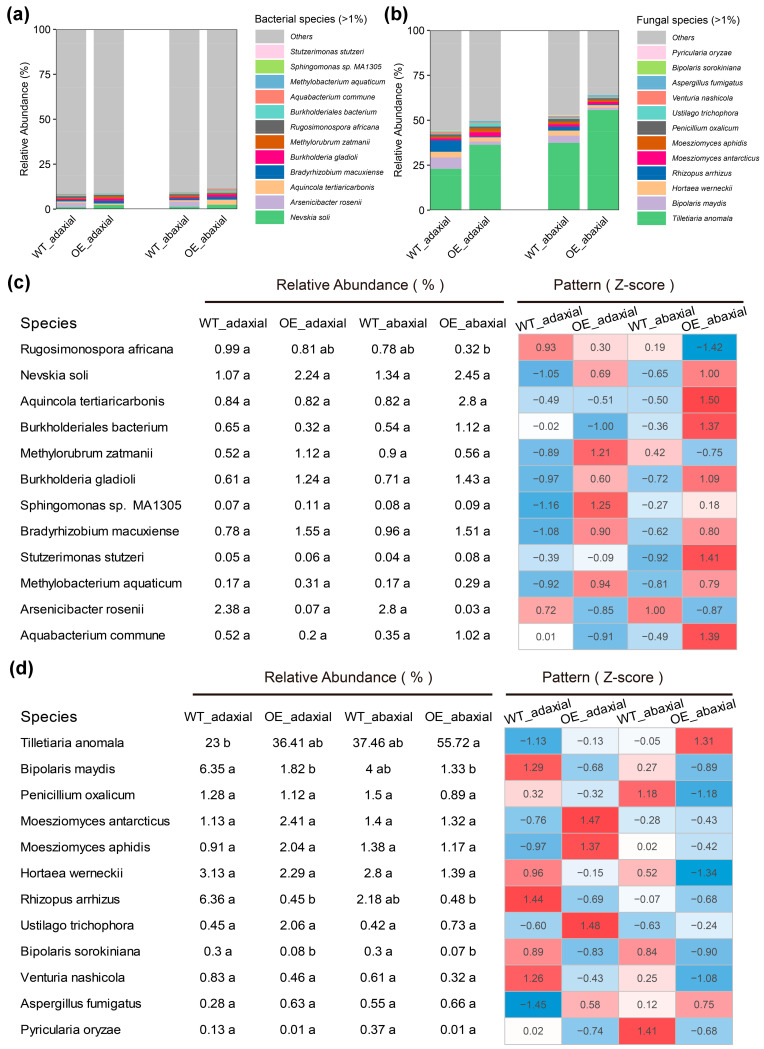
Relative abundances of dominant bacterial and fungal species in maize. (**a**) Bacterial and (**b**) fungal species with relative abundances > 0.1%. To visualize variation patterns, (**c**) Bacterial and (**d**) fungal species with relative abundances were normalized to Z-score values, where Z-score = (data point − mean)/(standard deviation). WT-adaxial, adaxial surfaces of leaves from wild-type maize; OE-adaxial, adaxial surfaces of leaves from *ZmMYB3R*-overexpressing maize; WT-abaxial, abaxial surfaces of leaves from wild-type maize; OE-abaxial, abaxial surfaces of leaves from *ZmMYB3R*-overexpressing maize. Different letters indicate significant differences between niches (*n* = 3, *p* < 0.001, analysis of variance (ANOVA)).

**Figure 4 microorganisms-13-00362-f004:**
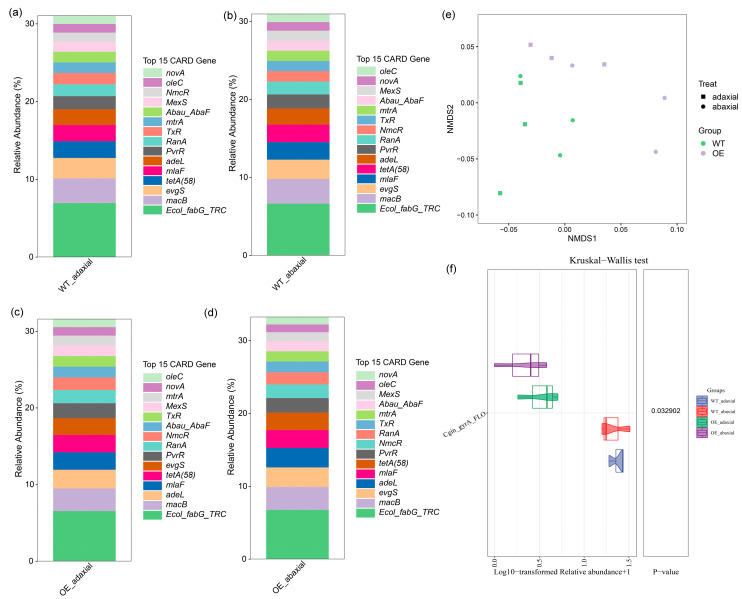
Dominant gene categories and differences between the dominant genes on the adaxial and abaxial surfaces of leaves of wild-type (WT) and transgenic (OE) maize. (**a**–**d**) Bar plot showing the average relative abundances of the top 15 functional Comprehensive Antibiotic Resistance Database (CARD) genes detected in all phyllosphere samples. (**e**) Variation in phyllosphere functional gene profiles observed in this study. A non-metric multi-dimensional scaling (NMDS) plot of Bray–Curtis similarities calculated based on the relative abundance of SEED subsystem level 3 functional CARD genes. (**f**) Phyllosphere microbial functional gene that significantly (ANOVA, *p* < 0.05) and consistently differed in relative abundance for each surface of WT and OE. WT-adaxial, adaxial surfaces of leaves from wild maize; OE-adaxial, adaxial surfaces of leaves from transgenic maize (*ZmMYB3R*-OE); WT-abaxial, abaxial surfaces of leaves from wild maize; OE-abaxial, abaxial surfaces of leaves from *ZmMYB3R*-OE.

**Figure 5 microorganisms-13-00362-f005:**
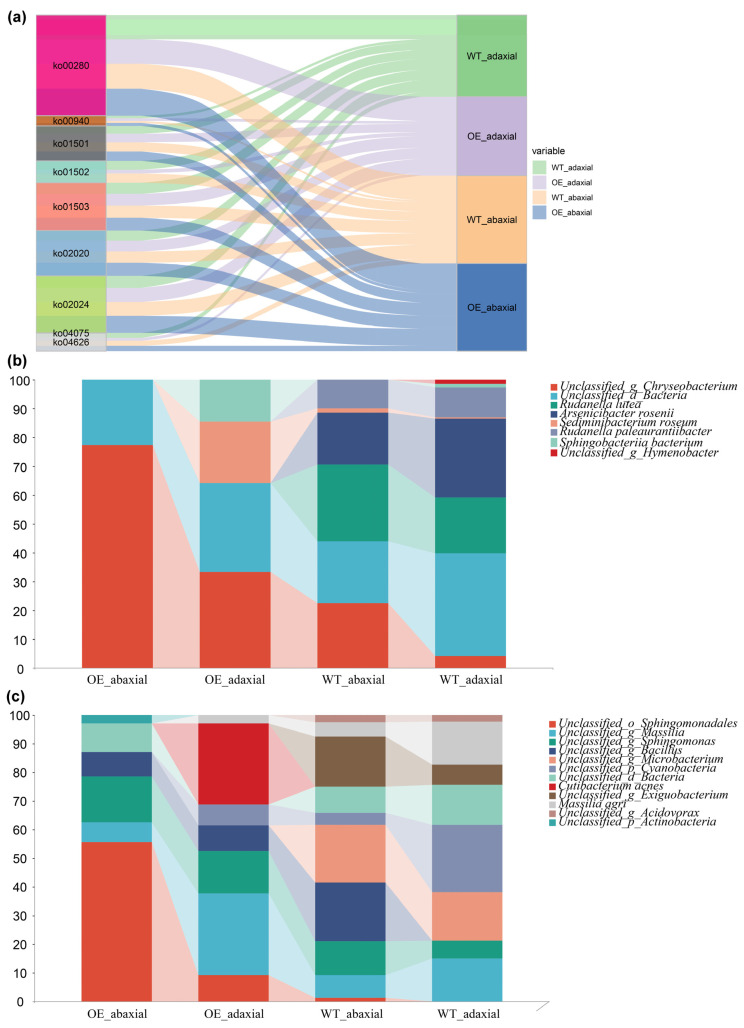
Functional genes and their host microbiome in the maize phyllosphere. (**a**) Sankey diagram reflecting the abundance of microorganisms involved in disease resistance-related pathway metabolism in wild-type (WT) maize. (**b**) Proportion of *Cgin_gyrA_FLO* in the microbial community. (**c**) Proportion of *mecA* in the microbial community. WT-adaxial, adaxial surfaces of leaves from wild-type maize; OE-adaxial, adaxial surfaces of leaves from *ZmMYB3R*-overexpressing maize; WT-abaxial, abaxial surfaces of leaves from wild-type maize; OE-abaxial, abaxial surfaces of leaves from *ZmMYB3R*-overexpressing maize. ko00280, aromatic compound metabolism; ko00940, phenylpropanoid biosynthesis; ko01501, beta-lactam resistance; ko01502, vancomycin resistance; ko01503, cationic antimicrobial peptide (CAMP) resistance; ko02020, two-component system; ko02024, quorum sensing; ko04075, plant hormone signal transduction (jasmonic acid and ethylene signaling pathway); ko04626, plant–pathogen interaction.

**Figure 6 microorganisms-13-00362-f006:**
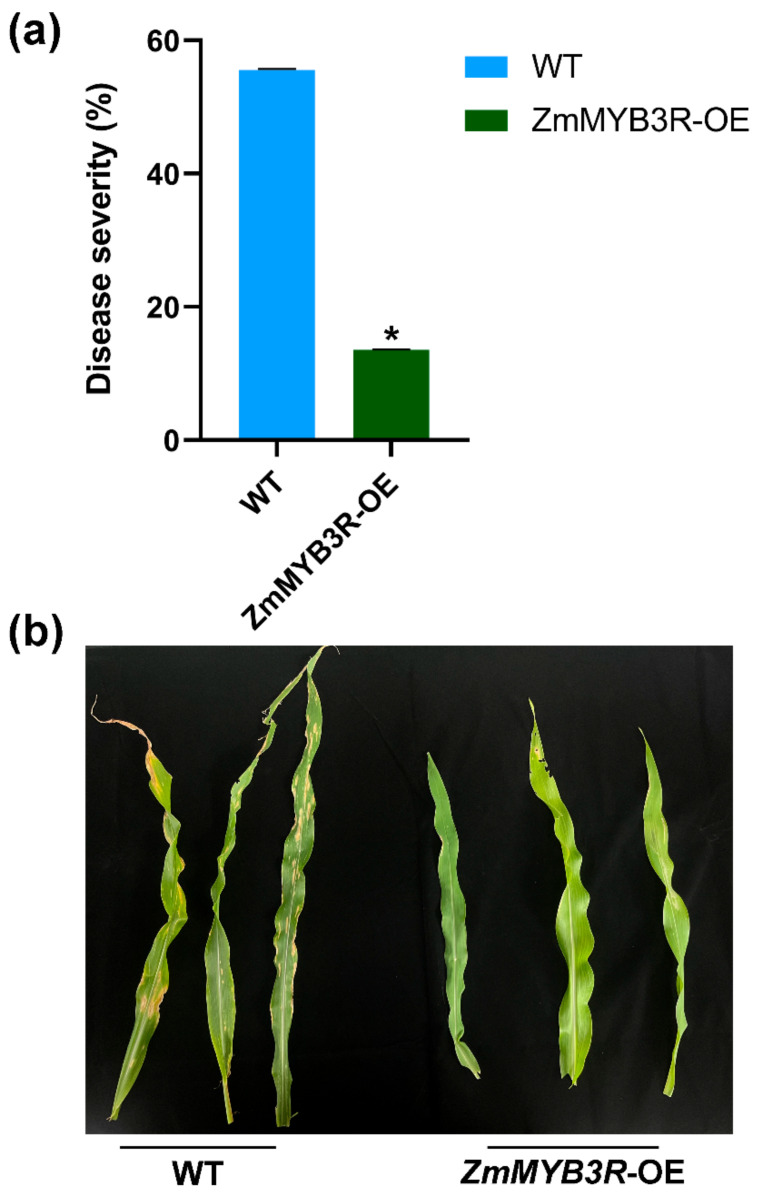
Disease severity and symptomology of *Exserohilum turcicum* leaf infection on maize. (**a**) Disease severity of WT and ZmMYB3R-OE after inoculation of *Exserohilum turcicum*. (**b**) Symptom of WT and ZmMYB3R-OE after inoculation of *Exserohilum turcicum*. Disease severity was evaluated 14 days post inoculation (* *p* ≤ 0.05).

**Table 1 microorganisms-13-00362-t001:** Linear-mixed models (LMMs) for alpha diversity indices. The effects of the genotype and niche on bacterial and fungal community alpha diversity indices were tested with LMMs. Significance was assessed using type II analysis of variance (ANOVA) with Kenward–Rodger approximation of the degrees of freedom in an LMM.

Microbial Communities	Variables	Shannon Diversity		Chao1 Index	
		*F* Value	*p* (>*F*)	*F* Value	*p* (>*F*)
Bacterial Community	Genotype	9.329	0.014	19.595	0.002
	Niche	2.281	0.165	2.343	0.160
Fungal Community	Genotype	9.698	0.012	18.246	0.002
	Niche	6.013	0.037	13.613	0.005

**Table 2 microorganisms-13-00362-t002:** Effects of maize genotypes and niche on microbial community structure assessed by permutational analysis of variance (PERMANOVA).

Microbial Community	Factor	*F* Value	R Square	*p* Value
Bacterial Community	Niche	0.6515	0.06116	0.717
	Genotype	3.4922	0.25883	0.005 **
	Niche*Genotype	1.6014	0.37521	0.085
	Pairwise comparison			
	OE-abaxial vs. OE-adaxial	1.17147	0.22653	0.3
	WT-abaxial vs. WT-adaxial	0.52571	0.11616	0.7
	WT-adaxial vs. OE-adaxial	1.91423	0.32366	0.2
	WT-abaxial vs. OE-abaxial	2.11042	0.34538	0.2
Fungal Community	niche	2.7998	0.21874	0.059
	Genotype	4.5561	0.313	0.012 *
	niche*Genotype	3.8189	0.58883	0.003 **
	Pairwise comparison			
	OE-abax ial vs. OE-adaxial	2.21219	0.3561	0.2
	WT-abaxial vs. WT-adaxial	4.02544	0.50158	0.1
	WT-adaxial vs. OE-adaxial	3.89427	0.4933	0.1
	WT-abaxial vs. OE-abaxial	3.154	0.44087	0.2

Niche*Genotype represent the interactive effects of niche and genotype. Significant differences are marked with asterisks: *, *p* < 0.05; **, *p* < 0.01.

**Table 3 microorganisms-13-00362-t003:** Topological characteristics of phyllosphere bacterial–fungal networks.

Network Indicators	WT-Adaxial	OE-Adaxial	WT-Abaxial	OE-Abaxial
Number of nodes	100	100	99	98
Ratio of fungal/bacterial nodes	1	1	0.98	0.96
Number of edges	862	605	843	798
Number of positive correlations	412	298	432	311
Number of negative correlations	450	307	411	487
Average length	1.493537015	1.629061925	1.491862568	1.512820513
Graph density	0.174141414	0.122222222	0.173778602	0.167893962
Network diameter	2	2	2	2
Clustering coefficient	0	0	0	0
Average degree	17.8	21.52	17.48484848	18.14285714
Modularity	0.595563654	0.577304829	0.615949646	0.59595103
Neg/pos	1.09223301	1.030201342	0.951388889	1.565916399

## Data Availability

Data will be made available upon request.

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
