# Peer review of "Transgenic Maize of ZmMYB3R Shapes Microbiome on Adaxial and Abaxial Surface of Leaves to Promote Disease Resistance"

_microorganisms, 2025, doi:10.3390/microorganisms13020362_

Round 1
Reviewer 1 Report
Comments and Suggestions for Authors
The manuscript, titled "Transgenic maize of ZmMYB3R shapes microbiome on adaxial and abaxial surface of leaves to promote disease resistance," sheds light on an interesting aspect of transgenic plants. Transformation with a gene that enhances drought tolerance not only changes the agronomic properties of the crop, but also results in changes in the microbial communities that develop on the leaves.
Moreover, these changes can be completely different depending on whether the leaf upper or the lowerside is examined. In the presented studies, microbial communities of a transgenic and a "wild" corn type were compared with metagenomic sequencing.
Individual testing methods and statistical analyses, as well as artificial infection studies, have also demonstrated that changes in the composition of microbial communities can result in increased disease tolerance in the genetically modified maize type.
The manuscript reflects research work compiled with due care and detail. The results are certainly noteworthy for further research. The presentation of the results is sufficiently detailed, the analyses are versatile.
I recommend publication in the form of a scientific article.
Author Response
Comments:
The manuscript, titled "Transgenic maize of ZmMYB3R shapes microbiome on adaxial and abaxial surface of leaves to promote disease resistance," sheds light on an interesting aspect of transgenic plants. Transformation with a gene that enhances drought tolerance not only changes the agronomic properties of the crop, but also results in changes in the microbial communities that develop on the leaves.
Moreover, these changes can be completely different depending on whether the leaf upper or the lowerside is examined. In the presented studies, microbial communities of a transgenic and a "wild" corn type were compared with metagenomic sequencing.
Individual testing methods and statistical analyses, as well as artificial infection studies, have also demonstrated that changes in the composition of microbial communities can result in increased disease tolerance in the genetically modified maize type.
The manuscript reflects research work compiled with due care and detail. The results are certainly noteworthy for further research. The presentation of the results is sufficiently detailed, the analyses are versatile.
I recommend publication in the form of a scientific article.
Response:
Thanks for the kind and insightful comments. The unexpected effects brought by transgene are not only of great significance for safety evaluation, but also provide a theoretical basis for the development of functional microorganisms and the study of the correlation between microorganisms and different genotypes of crops.

Reviewer 2 Report
Comments and Suggestions for Authors
Dear Authors,
below are some comments on the manuscript you submitted. Please respond in detail to these few questions/suggestions.
1. The study noted that ZmMYB3R overexpression affected the diversity of the microbiome (Chao1 index). Could the authors explain what biological mechanisms might be behind this phenomenon? How might specific changes in the composition of the microbiome contribute to the observed improvement in disease resistance?
2. Changes in the microbiome were studied on both leaf surfaces (adaxial and abaxial). What specific differences in the microbial environment between the two surfaces might affect the structure of their microbiome? Are there data on differences in water or nutrient availability that could explain these observations?
3. The authors highlight that the ZmMYB3R transgene may have altered the balance between commensal and pathogenic microorganisms. Could they provide additional evidence for this interaction? What specific metabolic pathways or evolutionary mechanisms could be responsible for changes in this balancing?
4. Sampling methodology: The methodology describes how to collect microbiome samples from the leaf surface. Could the authors clarify whether uniform procedures were used to collect samples from both adaxial and abaxial surfaces?
5. The article mentions potential applications of the results in managing the leaf microbiome for better disease resistance. What specific recommendations or strategies could you suggest for corn growers based on the results of this study? Is there further research planned to test the effectiveness of these approaches in the field?
Author Response
Comments:
Dear Authors,
below are some comments on the manuscript you submitted. Please respond in detail to these few questions/suggestions.
Comments 1: The study noted that ZmMYB3R overexpression affected the diversity of the microbiome (Chao1 index). Could the authors explain what biological mechanisms might be behind this phenomenon? How might specific changes in the composition of the microbiome contribute to the observed improvement in disease resistance?
Response 1: Thank you for pointing this out.
The Chao1 index is a commonly used indicator to measure microbial diversity, particularly for estimating species richness. Research has shown that transgenic crops can alter the composition and diversity of microbial communities through various mechanisms, which may have an impact on host health, particularly in terms of disease resistance.
Transgenic plants mainly affected microbial communities through two pathways. One is the expression products of exogenous genes from transgenic plants have a certain impact on the diversity of microbiome communities, such as the secretion of exogenous proteins into the soil by the residues or roots of transgenic plants, the other is the horizontal gene transfer of exogenous genes of transgenic plants, that is, the integration of DNA into the genome of microbiome by transferring it to other cells without offspring, which may change the genetic characteristics and functions of microbiome (Dunfield & Germida 2004; O'Donnell & Gorres 1999).
Transgenic Arabidopsis overexpressing ZmMYB3R displayed enhanced growth performance and higher survival rates, elevated catalase (CAT), peroxidase (POD), superoxide dismutase (SOD) enzyme activities and ABA content, increased sensitivity to ABA, and regulation of the stomatal aperture (Wu et al., 2019). So we speculated overexpression of ZmMYB3R may indirectly affect the structure of the microbial community by regulating plant secondary metabolites or exudates, which may have a selective effect on the microbial community and alter microbial diversity.
Previous studies have shown that changes in the composition of the microbiome may improve disease resistance by enriching beneficial microorganisms and enhancing the plant's immune system (Berg 2009; Bulgarelli et al., 2013). Transgenic crops may promote the growth of beneficial microorganisms by changing the chemical composition of the soil or rhizosphere environment. These microorganisms can promote plant health through a variety of mechanisms, including directly inhibiting the growth of pathogenic microorganisms and activating the plant's immune system. Some transgenic crops enhance the plant's immune system through the expression of disease-resistant genes. And the activation of these immune systems may further promote interactions with microorganisms and produce a richer microbial community.
Our study indeed confirmed that more beneficial microorganisms accumulated in transgenic plants, such as Burkholderiales bacterium, Burkholderia gladioli, Methylorubrum zatmanii, Methylobacterium aquaticum, and Sphingomonas sp. MA1305, and studies revealed that their functions related to promoting plant growth or enhancing plant disease resistance. Relevant content has been added to the discussion. The levels of kinases and hormones related to stress have also increased in transgenic Arabidopsis (Wu et al., 2019), these results collectively support our speculation that ZmMYB3R may enhance plant disease resistance by regulating phyllosphere microbiome.
Please see our revised manuscript for details (Page 14, line 433-460.)
References:
Dunfield K E, Germida J J.Impact of Genetically Modified Crops on Soil- and Plant-Associated Microbial Communities [J].Journal of Environmental Quality, 2004, 33(3). doi:10.2134/jeq2004.0806.
O'Donnell A G, Gorres H E.16S rDNA methods in soil microbiology [J].Current Opinion in Biotechnology, 1999, 10(3):225-229. doi:10.1016/S0958-1669(99)80039-1.
Wu J, Jiang Y, Liang Y, Chen L, Chen W, Cheng B. Expression of the maize MYB transcription factor ZmMYB3R enhances drought and salt stress tolerance in transgenic plants. Plant Physiol Biochem. 2019 Apr;137:179-188. doi: 10.1016/j.plaphy.2019.02.010.
Berg G. Plant–microbe interactions promoting plant growth and health: perspectives for controlled use of microorganisms in agriculture[J].Applied Microbiology & Biotechnology, 2009, 84(1):11-18.doi:10.1046/j.1439-037x.1999.00294.x.
Bulgarelli D, Schlaeppi K, Spaepen S, Ver Loren van Themaat E, Schulze-Lefert P. Structure and functions of the bacterial microbiota of plants. Annu Rev Plant Biol. 2013; 64:807-38. doi: 10.1146/annurev-arplant-050312-120106.
Comments 2: Changes in the microbiome were studied on both leaf surfaces (adaxial and abaxial). What specific differences in the microbial environment between the two surfaces might affect the structure of their microbiome? Are there data on differences in water or nutrient availability that could explain these observations?
Response 2: Thanks for the kind suggestions.
The differences in microenvironmental conditions faced by the two leaf surfaces may also affect their infection phenotype. Temperature, humidity, nutrient, and CO2 all affect the composition of microbial communities (Zhan et al., 2022). The temperature and light intensity differences between the two surfaces of leaves may also affect the severity of diseases caused by bacterial and fungal pathogens (Wu et al., 2024). There may be anatomical differences in stomatal density on the two surface of leaves, the stomata number, stomata index and stomata length increased in the upper surfaces of leaves in comparison with their lower surfaces (KürÅŸat et al., 2007), and stomatal density is positively correlated with the development of diseases (Gupt et al., 2021), the local relative humidity on the abaxial of leaves is usually higher than that on the adaxial, which may promote the occurrence of downy mildew caused by oomycete pathogens (Purayannur et al., 2021; Nirwan et al., 2023). Previous studies have shown that ZmMYB3R enhanced drought resistance by controlling stomatal closure (Wu et al., 2019). The opening and closing of stomata can control the gas exchange and water evaporation between plants and the outside world.
Please see our revised manuscript for details (Page 14-15, line 475-478, line 479, line 481-488.)
We did not specifically measure the differences in moisture or nutrients between the adaxial and abaxial surfaces of the leaves, based on metagenomic sequencing results and previous studies, we speculate that changes in the microbial community are due to overexpression of ZmMYB3R.
References:
Gupt, S. K., Chand, R., Mishra, V. K., Ahirwar, R. N., Bhatta, M., & Joshi, A. K. Spot blotch disease of wheat as influenced by foliar trichome and stomata density.Journal of Agriculture and Food Research, 2021, 6. doi:10.1016/j.jafr.2021.100227.
KürÅŸat ÇavuÅŸoÄŸlu. Effects of pretreatments of some growth regulators on the stomata movements of barley seedlings grown under saline (NaCl) conditions. Plant Soil & Environment, 2007, 53(12):524-528. doi:10.1111/j.1365-3059.2007.01733.x.
Nirwan S, Sharma AK, Tripathi RM, Pati AM, Shrivastava N. Resistance strategies for defense against Albugo candida causing white rust disease. Microbiol Res. 2023 May; 270:127317. doi: 10.1016/j.micres.2023.127317.
Pruitt RN, Locci F, Wanke F, Zhang L, Saile SC, Joe A, Karelina D, Hua C, Fröhlich K, Wan WL, Hu M, Rao S, Stolze SC, Harzen A, Gust AA, Harter K, Joosten MHAJ, Thomma BPHJ, Zhou JM, Dangl JL, Weigel D, Nakagami H, Oecking C, Kasmi FE, Parker JE, Nürnberger T. The EDS1-PAD4-ADR1 node mediates Arabidopsis pattern-triggered immunity. Nature. 2021 Oct; 598(7881):495-499. doi: 10.1038/s41586-021-03829-0.
Wu J, Jiang Y, Liang Y, Chen L, Chen W, Cheng B. Expression of the maize MYB transcription factor ZmMYB3R enhances drought and salt stress tolerance in transgenic plants. Plant Physiol Biochem. 2019 Apr;137:179-188. doi: 10.1016/j.plaphy.2019.02.010.
Wu Y, Sexton WK, Zhang Q, Bloodgood D, Wu Y, Hooks C, Coker F, Vasquez A, Wei CI, Xiao S. Leaf abaxial immunity to powdery mildew in Arabidopsis is conferred by multiple defense mechanisms. J Exp Bot. 2024 Feb 28;75(5):1465-1478. doi: 10.1093/jxb/erad450.
Zhan C, Matsumoto H, Liu Y, Wang M. Pathways to engineering the phyllosphere microbiome for sustainable crop production. Nat Food. 2022 Dec;3(12): 997-1004. doi: 10.1038/s43016-022-00636-2.
Comments 3: The authors highlight that the ZmMYB3R transgene may have altered the balance between commensal and pathogenic microorganisms. Could they provide additional evidence for this interaction? What specific metabolic pathways or evolutionary mechanisms could be responsible for changes in this balancing?
Response 3: Thanks for your careful comments.
In this study, we found more beneficial microbes accumulated on the phyllosphere of transgenic maize, while pathogenic ones diminished, this lead us to speculate that the disease resistance of ZmMYB3R-OE has changed, and we only suspect that the balance between symbiotic microorganisms and pathogenic microorganisms has been disrupted. We don't have any additional direct evidence to prove this speculation, more direct evidence requires generating relevant transgenic materials in future experiments, such as knocking out the ZmMYB3R in maize, comparing the phyllosphere microbiome of transgenic maize that overexpressed and knocked out ZmMYB3R, and analyzing the differences between the them.
The metagenomic sequencing results revealed functional genes were enriched in nine disease-resistance pathways. They were involved in aromatic compound metabolism, phenylpropanoid biosynthesis, be-ta-lactam resistance, vancomycin resistance, cationic antimicrobial peptide (CAMP) resistance, two-component system, quorum sensing, plant hormone signal transduction (jasmonic acid and ethylene signaling pathway), and plant-pathogen interaction. The relative abundances of most resistance-related metabolic pathways in the OE-abaxial community were higher than in the WT-adaxial, WT-abaxial, and OE-adaxial communities (Figure S5). According to previous studies, changes in plant metabolic pathways and microbial metabolic pathways may lead to changes in the balance between symbiotic microorganisms and pathogenic microorganisms. The secretion of secondary metabolites, such as phenolic acids secreted by plants, has antibacterial effects and can inhibit the reproduction of certain pathogenic microorganisms (Berendsen et al., 2012). Sugars and amino acids secreted by plants can provide nutrients for nitrogen-fixing bacteria or growth-promoting bacteria, helping them to colonize in the rhizosphere (Berendsen et al., 2012). Plant hormones not only regulate the growth of plants themselves, but also affect their interactions with plants by regulating the composition of microbial communities. Auxins and auxin-producing bacteria can promote the production of antioxidant compounds and osmotic regulators in plants, which can counteract cadmium toxicity, promote plant growth, and improve plant remediation efficiency (Rolón-Cárdenas et al., 2022). Auxins, cytokinin and ethylene can affect rhizosphere microorganisms and induce systemic drought tolerance in plants (Nio et al., 2023). ABA affected the rhizosphere bacterial community during the later developmental stages, thereby further influencing the composition of rhizobia by affecting compounds in root exudates (Lopes et al., 2023). ZmMYB3R has been shown to enhanced drought resistance by controlling increase stomatal closure and ABA content in Arabidopsis (Wu et al., 2019).
The evolutionary trajectories of symbiotic and pathogenic interactions are constrained and interconnected. Symbiotic microorganisms usually inhibit the growth of pathogenic microorganisms by synthesizing antibiotics, enzymes or toxic metabolites (Delaux et al., 2021). Changes in the host immune system (such as by initiating innate immune responses, changing the distribution or function of immune cells, etc.) may affect the competition and symbiotic relationship between symbiotic microorganisms and pathogenic microorganisms, thereby affecting host health and disease resistance (Berendsen et al., 2012).
Please see our revised manuscript for details (Page 15, line 493-519.)
References:
Berendsen RL, Pieterse CM, Bakker PA. The rhizosphere microbiome and plant health. Trends Plant Sci. 2012 Aug;17(8):478-86. doi: 10.1016/j.tplants.2012.04.001.
Rolón-Cárdenas GA, Arvizu-Gómez JL, Soria-Guerra RE, Pacheco-Aguilar JR, Alatorre-Cobos F, Hernández-Morales A. The role of auxins and auxin-producing bacteria in the tolerance and accumulation of cadmium by plants. Environ Geochem Health. 2022 Nov;44(11):3743-3764. doi: 10.1007/s10653-021-01179-4.
Nio SA, Mantilen Ludong DP. Beneficial Root-Associated Microbiome during Drought and Flooding Stress in Plants. Pak J Biol Sci. 2023 Apr;26(5):287-299. doi: 10.3923/pjbs.2023.287.299
Lopes LD, Futrell SL, Bergmeyer E, Hao J, Schachtman DP. Root exudate concentrations of indole-3-acetic acid (IAA) and abscisic acid (ABA) affect maize rhizobacterial communities at specific developmental stages. FEMS Microbiol Ecol. 2023 Feb 28;99(3):fiad019. doi: 10.1093/femsec/fiad019.
Delaux PM, Schornack S. Plant evolution driven by interactions with symbiotic and pathogenic microbes. Science. 2021 Feb 19;371(6531):eaba6605. doi: 10.1126/science.aba6605.
Comments 4: Sampling methodology: The methodology describes how to collect microbiome samples from the leaf surface. Could the authors clarify whether uniform procedures were used to collect samples from both adaxial and abaxial surfaces?
Response 4: Thank you for pointing this out.
The sampling position on the adaxial surface of the leaves corresponded to the sampling position on the abaxial surface. As you suggested, the procedure of collect samples from both adaxial and abaxial surfaces was clearly rewritten for a better understanding of the details in this experiment.
Please see our revised manuscript for details (Page 3, and line 116-119).
Comments 5: The article mentions potential applications of the results in managing the leaf microbiome for better disease resistance. What specific recommendations or strategies could you suggest for corn growers based on the results of this study? Is there further research planned to test the effectiveness of these approaches in the field?
Response 5: Thanks for your careful comments.
As a transcription factor that regulated plant immunity and microbial community composition, ZmMYB3R might improve maize disease resistance and reduce disease losses by optimizing phyllosphere microbial communities.
There are some recommendations for planting strategies based on ZmMYB3R. Promote the use of maize varieties that overexpressing ZmMYB3R (ZmMYB3R-OE). Growers can consider cultivating ZmMYB3R-OE maize varieties. Overexpression of ZmMYB3R may help to regulate the microbial community on the phyllosphere, selectively promote the growth of beneficial microorganisms, and inhibit the expansion of pathogenic microorganisms. The transgenic maize enhanced maize immune response, especially against common diseases (such as NCLB (Northern corn leaf blight), maize leaf spot, etc.), thereby improving maize disease resistance.
Combined application of plant protection agents. Based on overexpression of ZmMYB3R in maize, growers can use biological control methods, such as probiotics or biopesticides to further enhance the health of leaf microbial communities. For example, the use of products containing probiotic microorganisms (such as Bacillus subtilis and Trichoderma spp.) can enhance the disease resistance of ZmMYB3R-OE maize. These microorganisms can not only inhibit pathogenic microorganisms, but also establish a symbiotic relationship with plants, thereby improving the stress resistance of plants.
Agronomic management measures. Based on the impact of ZmMYB3R on the microbial community, maize growers can take the following measures to further optimize the microbial environment of plants. Improve soil health and promote the colonization of beneficial microorganisms by increasing the content of soil organic matter and adjusting soil pH. Reasonable irrigation and fertilization can maintain the stability of phyllosphere and rhizosphere microbial communities and avoid overfertilization and waterlogging, which may affect the balance of microorganisms and thus affect disease resistance.
Mixed cropping and rotation. Crop rotation and mixed cropping are also effective strategies to increase the diversity of microbial communities and improve disease resistance. When planting maize, it can be rotated with other crops (such as beans and wheat) to change the structure of the microbial community, reduce the accumulation of soil pathogens, and promote the reproduction of beneficial microorganisms in the soil, thereby indirectly improving the disease resistance of maize.
In order to verify the actual effect of ZmMYB3R in phyllosphere microbiome management and ensure its application effect in the field, the following are research plans.
Field experiments. Conduct field trials of ZmMYB3R-OE maize to compare the performance of different maize varieties (including ZmMYB3R-OE varieties and conventional varieties) under natural disease pressure.
Microbiome monitoring. Analyze the differences in phyllosphere and rhizosphere microbial communities between ZmMYB3R-OE maize and conventional maize through metagenomics sequencing technology to verify the regulatory effect of ZmMYB3R on the microbial community in field.
Disease resistance evaluation: Record the growth of maize under different disease pressures (such as NCLB, maize leaf spot, gray leaf spot, etc.) to evaluate the improvement of disease resistance by overexpressing ZmMYB3R.
Microbiome intervention experiment. In ZmMYB3R-OE maize, analyze the role of different microbial communities on plant disease resistance. By applying specific probiotics (e.g., mixture of probiotics) to transgenic maize, test the effect of microbiome intervention on plant health and evaluate the improvement effect of different microbial community combinations on disease resistance.
Long-term stability study. In order to ensure the sustained effect of ZmMYB3R on maize disease resistance, long-term stability studies are needed, especially whether the effects of overexpressed ZmMYB3R on microbial communities and disease resistance are stable under different environmental conditions and agronomic management measures. These studies help to understand the adaptability of overexpressing ZmMYB3R to maize growth in different geographical regions and climatic conditions.
Study on the combined effect with other disease resistance mechanisms. ZmMYB3R may work together with other plant immune mechanisms (such as MAPK signaling pathway, NPR1-dependent immune response, etc.) to further enhance disease resistance. Therefore, the combined study of the synergistic effect of ZmMYB3R-OE and other disease resistance mechanisms can provide a more comprehensive strategy for maize disease resistance breeding.
Please see our revised manuscript for details (Page 16, and line 557-573).

Round 2
Reviewer 2 Report
Comments and Suggestions for Authors
I accept for publication